# Nanoparticle-based hollow microstructures formed by two-stage nematic nucleation and phase separation

Sheida T. Riahinasab[1], Amir Keshavarz[2], Charles N. Melton[1], Ahmed Elbaradei[1], Gabrielle I. Warren[2], Robin L.B. Selinger [3], Benjamin J. Stokes [2] & Linda S. Hirst [1]

Rapid bulk assembly of nanoparticles into microstructures is challenging, but highly desirable for applications in controlled release, catalysis, and sensing. We report a method to form hollow microstructures via a two-stage nematic nucleation process, generating size-tunable closed-cell foams, spherical shells, and tubular networks composed of closely packed nanoparticles. Mesogen-modified nanoparticles are dispersed in liquid crystal above the nematic-isotropic transition temperature ($T_{NI}$). On cooling through $T_{NI}$, nanoparticles first segregate into shrinking isotropic domains where they locally depress the transition temperature. On further cooling, nematic domains nucleate inside the nanoparticle-rich isotropic domains, driving formation of hollow nanoparticle assemblies. Structural differentiation is controlled by nanoparticle density and cooling rate. Cahn-Hilliard simulations of phase separation in liquid crystal demonstrate qualitatively that partitioning of nanoparticles into isolated domains is strongly affected by cooling rate, supporting experimental observations that cooling rate controls aggregate size. Microscopy suggests the number and size of internal voids is controlled by second-stage nucleation.

[1] Department of Physics, School of Natural Sciences, University of California, Merced, Merced, CA 95343, USA. [2] Department of Chemistry & Chemical Biology, School of Natural Sciences, University of California, Merced, Merced, CA 95343, USA. [3] Department of Physics, Kent State University, Kent, OH 44240, USA. Correspondence and requests for materials should be addressed to L.S.H. (email: lhirst@ucmerced.edu)

Materials with hollow microstructures such as spherical shells, networks, and tubes have many useful technological applications in areas such as catalysis, sensing, batteries, and encapsulation/controlled release[1,2]. Top-down synthetic strategies to produce hollow microstructures include the use of soft or hard templates[3], spray techniques[4], and microfluidic methods[5]. Hollow structures can also be formed via template-free self-assembly. A popular one-pot synthesis technique takes advantage of Ostwald ripening[6], where crystals initially nucleate as solid spheres arranged in a porous, polycrystalline texture, and then subsequently become hollow as smaller grains in the interior dissolve and recrystallize to larger grains on the exterior, forming spherical shells or tubes. One drawback of this method is that Ostwald ripening is slow, and typically requires hours of processing time. We report a template-free, rapid synthetic method to produce hollow microstructures composed of nanoparticles that self-assemble in less than 1 s into tightly packed hollow spheres, foams, and tubular networks. Our method is based on the use of a liquid crystal solvent which undergoes a two-stage nucleation process on cooling through the isotropic–nematic phase transition.

Dispersion and controlled assembly of nanoparticles in a soft material (i.e., polymer or liquid crystal) can produce a diverse array of interesting structured materials. Unlike conventional liquids, soft phases with orientational order can organize nanoparticles by aggregation (e.g., at topological defects.) The resulting composite material may retain advantageous physical properties of the matrix (elasticity, birefringence, electro-optic actuation, etc.). Alternately, stable nanostructures can be harvested by removal from the host phase.

Liquid crystals (LCs) are optically anisotropic fluids in which the constituent molecules exhibit local orientational order. LCs are particularly useful for display and photonics applications, in particular because surface anchoring conditions and confinement can be used to manipulate global molecular orientation and produce macroscopic domains with a defined optic axis. When particles are dispersed into an aligned nematic liquid crystal phase, depending on surface anchoring conditions on the particle, an elastic deformation of the LC director may be imposed. Ligands can be used to define surface anchoring and force the surrounding LC molecules to align at a particular angle relative to the surface (perpendicular to a spherical particle for example). This means the inclusion of a particle creates spatial frustration, relaxed by the formation of topological defects. Recently, there has been much interest in nanoparticle and colloidal assembly at interfaces[7] and via topological defect lines and points in the nematic phase[8,9].

In recent years the field of soft nanocomposites has grown rapidly. Materials that combine nanoparticles with a fluid-like host show great potential for generation of soft-phase templated meta-materials[10–13] (e.g., biopolymers[14,15], biomolecules[16,17], or block copolymers[18–20]). These applications take advantage of a soft material's ability to spontaneously segregate and organize particles by their chemical and/or physical properties. Although soft host materials are complex fluids—intrinsically weakly ordered or disordered on the molecular scale—they often exhibit nano-to-micron-scale repeat units, as seen in the phase-separated microstructures of block copolymers[21], or the defect lattices of the LC blue phase[22]. Nanoparticle assembly can be achieved via particle patterning in topological defects or interfaces and many applications do not require a highly ordered particle lattice. Hence soft-phase assembly methods represent an attractive, fast, and low-cost approach for the production of interesting mesoscale hollow materials from nanoparticles.

Nanoparticle aggregation can also be driven by a phase transition in the host phase. In a series of seminal papers, Terentjev and colleagues[23–25] first reported the formation of micron-scale particle networks and cellular structures assembled at the isotropic-to-nematic phase transition using an elastically driven liquid crystal phase separation effect. They used the growth of ordered domains to generate a porous structure as colloidal particles were expelled from the nematic phase.

We expand on this pioneering work and use a dynamic two-stage nucleation and growth process to spatially organize nanoparticles, which are subsequently stabilized into a family of hollow structures. The method is rapid and template-free, performed close to room temperature using a widely available LC material, and, in principle, can be adapted to any nanoparticle type with appropriate surface modification. Instead of the micron-to-120 nm radius colloids used by Terentjev and colleagues[23–25], we report experiments using much smaller nanoparticles—6 nm diameter quantum dots. When dissolved in an LC host, such small particles depress the isotropic-to-nematic transition temperature[26] and exhibit high solubility in the isotropic phase but low solubility in the nematic phase. This combination of material properties gives rise to a two-stage nucleation process on cooling. The net Frank elastic energy cost for a particle to be located in the anisotropic nematic phase when compared to location in the isotropic phase provides an interesting mechanism for nanoparticle spatial organization[27–29]. When the temperature drops below the isotropic–nematic transition point, particles are expelled from nucleating nematic domains and segregate to isotropic domains. Due to the high nanoparticle (NP) concentration, these domains remain in the isotropic state as the isotropic–nematic transition temperature is locally depressed. On further cooling, the NP-rich isotropic domains undergo secondary nematic nucleation at a lower temperature. During this second stage of nucleation, NPs spontaneously segregate to the surface of the isotropic domains, thus forming hollow microstructures. This two-stage isotropic–nematic transition thus provides a mechanism to create hollow structures. To enable this process, our NPs are functionized with ligands selected to both promote solubility in the isotropic phase and enhance local attractive particle–particle interactions for final structure stability. Control over final morphology and pore size depends on the cooling rate though $T_{NI}$ and initial particle concentration in the liquid crystal solvent. The two-stage process is easier to control than the Ostwald ripening method, which depends on grain size distribution and diffusion rates. Besides proceeding much more rapidly, and providing morphological control by changing the NP density and cooling rate, one can also select a different LC solvent, or mixture of solvents, and/or change the species of ligands coating the NPs. These options open a broad chemical design space to optimize the process for any desired application.

## Results

**Formation of hollow nanoparticle-based microstructures**. The two basic elements of the liquid crystal/nanoparticle system used in this paper are a nematic liquid crystal (5CB, 4'-pentyl-4-biphenylcarbonitrile, Sigma Aldrich) and ligand-modified CdSe/ZnS core/shell quantum dots (LC-QDs) (NN Labs, core diameter 6.2 nm, absorption peak, $\lambda_{max}$ = 620 nm). The mesogenic ligand[30] **8**, inspired by the liquid crystals employed by the groups of Dunmur[31] and Vashchenko[32], was prepared using the sequence of reactions shown in Fig. 1d (see Supplementary notes III and IV for complete synthetic details).

The octadecylamine (ODA) surface ligands of the commercial QDs were exchanged with mesogenic ligand **8**. The mesogenic ligand's flexible amine tether is thought to encourage alignment with the local liquid crystal director 5CB, increasing dispersability

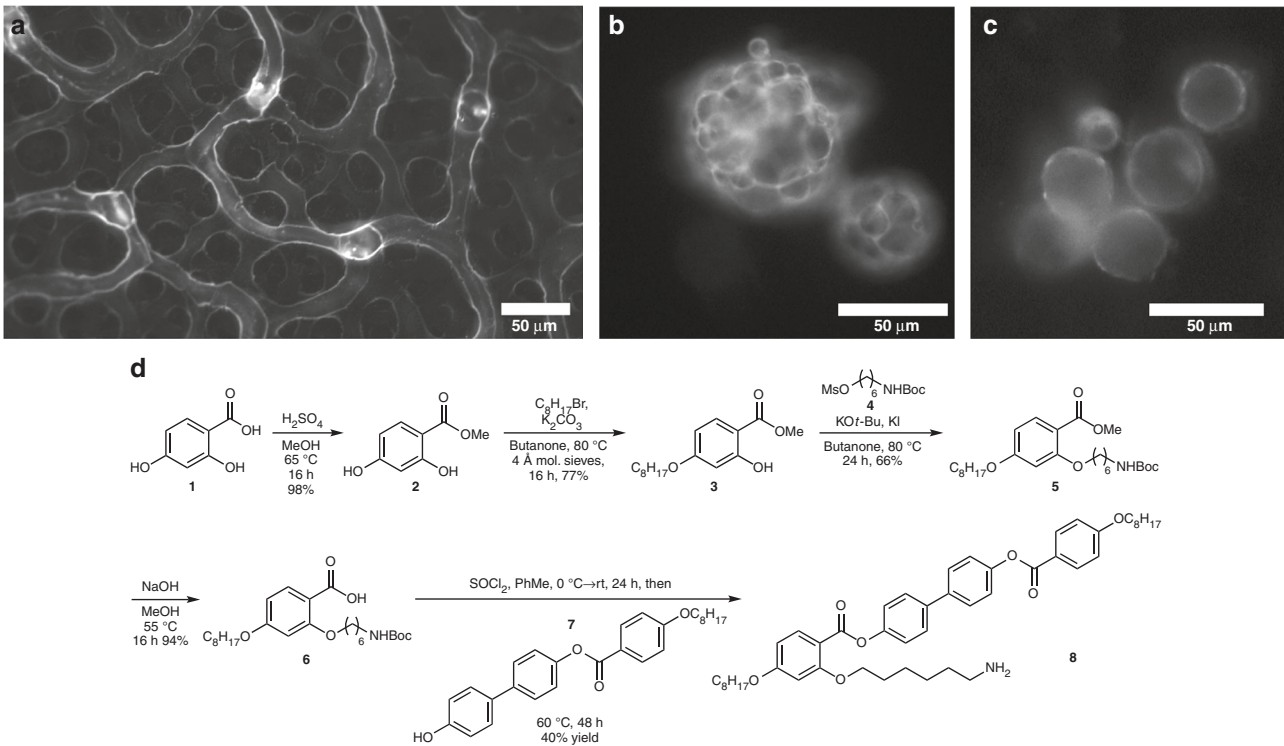

**Fig. 1** Hollow nanoparticle-based microstructures formed from ligand-modified quantum dots. Fluorescence microscopy imaging demonstrates the three distinct quantum dot structures formed under different conditions: **a** branching tubular network, formed at 1 °C min⁻¹, **b** solid closed-cell foam-like structures formed at 30 °C min⁻¹, and **c** hollow shell capsules formed at 200 °C min⁻¹. All structures are composed of 620 nm CdSe/ZnS ligand-modified quantum dots, suspended in nematic liquid crystal initially at 0.15 wt% at room temperature. **d** Sequence of reactions used to prepare the mesogenic ligand (**8**) for nanoparticle attachment

in the isotropic phase, while the rod-like aromatic motif may enable attractive interaction between closely packed particles[30]. The degree of ligand exchange was quantified using ¹H nuclear magnetic resonance (NMR) spectroscopy, revealing a 9:1 surface ratio of **8** to ODA (see Supplementary note V for full details).

Figure 1 shows representative fluorescence images of three distinct nanoparticle micro-morphologies: a branching network of tube-like structures (a); multi-compartment droplets of closed-cell foam (b); and single compartment hollow capsules (c). The structures are imaged suspended in nematic liquid crystal at room temperature. In the case of the individual capsules and the foam, liquid crystal is present throughout the structure (inside and outside the enclosed compartments), which we verified by cross-polarized microscopy. The final solid structures exhibit stable shapes—insensitive to thermal fluctuations—that are retained even if the host liquid crystal phase is heated above the isotropic transition point.

Qualitative observations initially revealed that final structure type and size could be controlled by varying either the initial particle concentration or the cooling rate through the isotropic-to-nematic transition. To examine these trends we constructed a qualitative morphological phase diagram as a function of these parameters, and also measured the size dependence of the final structures. In Fig. 2a–c, d–f, respectively, representative fluorescence images of the hollow shell morphology demonstrate the dependence of spherical shell and foam droplet size on cooling rate and concentration. To quantitatively analyze these data, we imaged a large number of shells and foam droplets. Figure 2h, i plots separately the outer dimensions of the assembled structures as a function of cooling rate and particle concentration. Structure diameters were measured using ImageJ software, and an average was taken over all structures formed under the same conditions.

For each data point shown on the graphs (Fig. 2h, i), the total number of measured structures, $n$, ranged from 10 to 183 (Supplementary Table 1). In some cases, low numbers were measured because a particularly large size (e.g., >50 μm in diameter at 7 °C min⁻¹, 0.3 wt%, Fig. 2a) produced fewer structures per microscope slide. In the case of multi-compartment foam droplets (Fig. 1b for example), the outer diameter was measured in several places, and an average value for each droplet was used (rather than the value of individual internal voids).

The diagram in Fig. 2g shows two important trends. Firstly, structural morphology is dependent on system cooling rate and, secondly, particle concentration is a factor in morphological size control. At high cooling rates (~200 °C min⁻¹), single hollow shells predominate, while at the lowest cooling rates, individual macrostructures do not stabilize, resulting in the network morphology. The intermediate range is particularly interesting: between cooling rates of 7 and 30 °C min⁻¹, we observed a surprising 'foam' structure: discrete compartmentalized 'droplets' suspended in the nematic host phase (Fig. 1b), with a closed-cell foam-like morphology. In the intermediate region of the diagram in Fig. 2g (indicated by the blue box), we often observed a mixture of individual capsules and foam-like structures (the colored boxes are a guide to indicate general behavior, not a discrete structural change). The network region was only observed at cooling rates below 7 °C min⁻¹; a characteristic lengthscale in that case was not measured.

Figure 3 shows more detailed confocal fluorescence imaging of the foam-like structure. In general, these multi-compartment structures form as discrete droplets (Fig. 3a–e), although more extended bulk foams, particularly near the edges of the formation chamber (Fig. 3f), are also observed. On first observation, foam

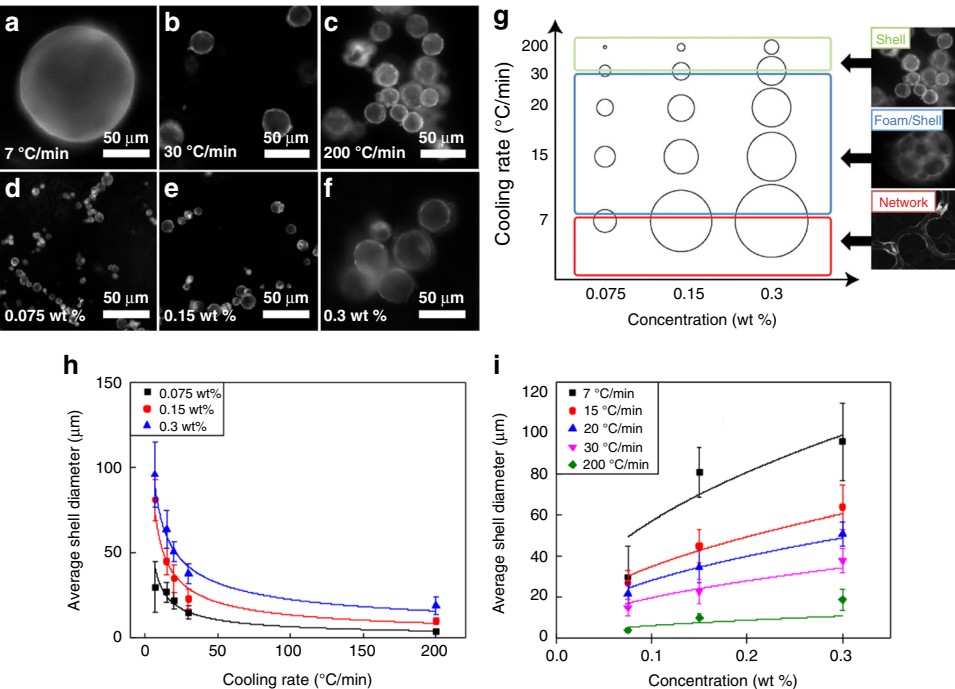

**Fig. 2** Size dependence on cooling rate and nanoparticle concentration. Representative fluorescence microscopy images of spherical shells formed from ligand-modified quantum dots (LC-QDs) at **a–c** fixed LC-QD concentration of 0.3 wt% in 5CB, varying cooling rate, and **d–f** fixed cooling rate of 200 °C min⁻¹, varying LC-QD concentration in 5CB. **g** Qualitative phase diagram for the three distinct morphologies predominantly observed as a function of cooling rate and concentration. General morphological zones on the diagram are indicated by the colored boxes with circle size representative of structure size. **h**, **i** Plots of average shell diameter vs. cooling rate and concentration respectively, with error bars indicating standard deviation (s.d.); the total number of measured structures, $n$, ranged from 10 to 183 (Supplementary Table 1)

droplets initially appeared to be composed of several shells fused together, but microscope observations of single compartment shells over time provided no evidence that fusion can take place after formation. The shells are mechanically quite rigid with walls that can be considered to be in a solid phase. Inner compartments were sometimes observed to merge during the formation process while the particles remained dispersed in the isotropic fluid (this phenomenon is shown in Supplementary Movie 1).

There is an interfacial tension between the nematic and isotropic phases, and as nematic domains shrink they tend to minimize their free energy by adopting minimal surface geometries—either spherical capsules or, in the case of the foam, a network of minimal interfaces. Our observations provide evidence that nanoparticles in shrinking domains remain in a dispersed fluid state until the point of arrest.

**Size dependence on cooling rate and nematic phase coarsening**. Figure 4a–d shows snapshots from a high-speed fluorescence movie (captured using a Phantom VEO-410L camera), recording the distribution of quantum dots in our composite material as the nematic phase nucleates, grows, and merges to leave shrinking isotropic domains (more detail can be seen in Supplementary Movies 1 and 2). Lighter regions indicate isotropic domains and dark regions highlight nematic domains, depleted of particles. The images in Fig. 4 highlight the stages of structure formation: initial phase separation (Fig. 4a), nematic domain growth (Fig. 4b), growth and merging of the nematic domains and transitioning to the point where the shrinking isotropic domains separate (Fig. 4c), and lastly shrinkage of those isotropic domains into capsules (Fig. 4d). This particular sequence leads to the formation of mostly single compartment shells. The time-point in Fig. 4c is notable as it indicates a stage where the isotropic

domains pinch off into droplets. These separating isotropic domains contain the particles that go on to form the final foams and capsules and we noted a qualitative relationship between early coarsening lengthscales and final capsule/foam size.

The process of initial nematic nucleation and growth can be described by a universal growth law, $L(t) \sim t^d$, where $t$ is the time after an initial temperature quench, $L$ is the domain diameter, and $d$ the growth exponent. Experimentally, this growth exponent, $d$, has been measured to lie between 0.5 and 1.0 for the I–N phase transition, depending on quench depth[33]. In a more recent numerical study, Bradač et al.[34] predicted the influence of quench time $t_q$ on early-stage I–N coarsening dynamics to be a power law, relating the average size of protodomains, $\xi$, to the quench time, in the form, $\xi \sim t_q^n$. When $\xi$ was measured at a fixed time for different quenches, the power law was obtained as $n = 0.25$. Quench time (i.e., time over which a quench crosses the transition) is inversely related to cooling rate, and hence we can expect the characteristic size of our assembled structure, $(\xi_r)$ to increase as cooling rate decreases. It is not obvious how these nucleation and growth models apply to our system where particle concentration is time dependent. In the initial stages of nematic nucleation and coarsening (wherein particle concentrations are still relatively low), however, we can focus on the dependence of domain lengthscales with cooling rate. Fitting a power law to the data shown in Fig. 2h produced a fit of the form $\xi_r \sim c^{-n}$, where $n = 0.51$, 0.63, and 0.69, and $c$ represents cooling rate, for three different particle concentrations, respectively. It is likely that larger shells arrest at an earlier time because they more rapidly reach maximum particle density at the shrinking spherical interface, given a constant cooling rate. A $y = x^{0.5}$ fit to the data in Fig. 2i supports this hypothesis.

To further investigate the scaling relationship between cooling rate and domain lengthscales, we carried out a simulation study

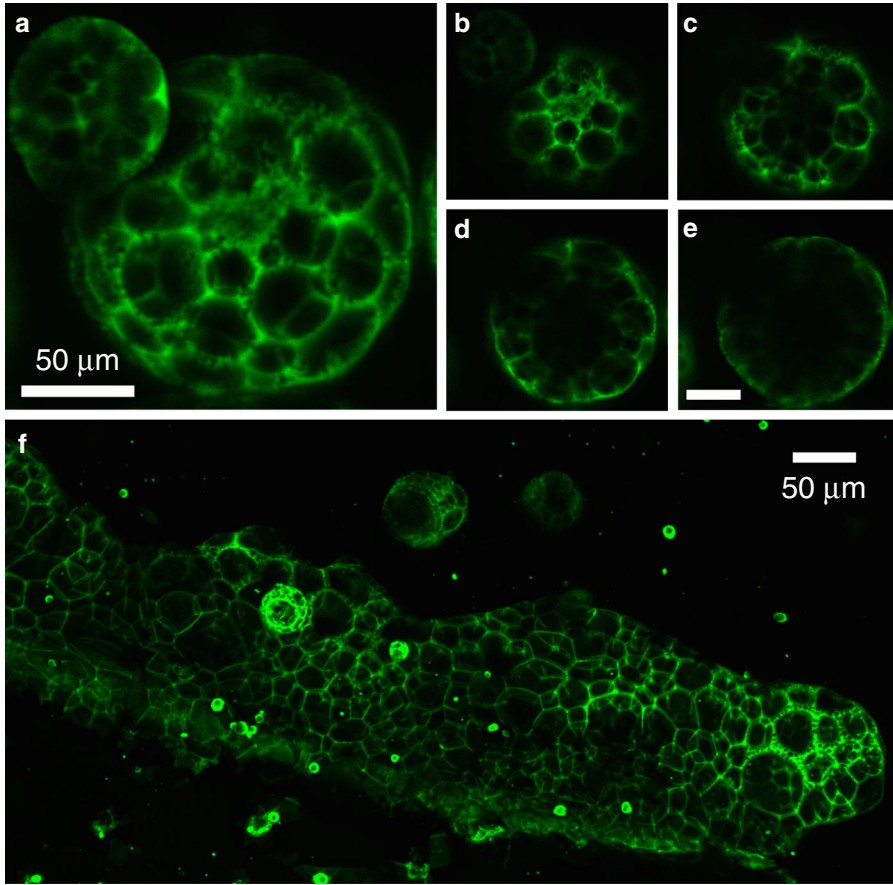

**Fig. 3** Confocal imaging of foam structures. Confocal microscope *z*-projection images of **a** quantum dot foam droplet (*z* depth = 23.13 μm) with four representative slices (**b**–**e**) at *z* intervals of 0.665 μm; and **f** a large foam structure (*z* depth = 91.90 μm). Scale bars for **b**–**e** are 50 μm

of the process by which phase separation (into nanoparticle-rich and nanoparticle-poor domains) is driven by the growth of the order parameter in the nematic phase. Our aim was to model only the first stage of the phase separation process and see if the lengthscale/cooling rate dependence could qualitatively be recovered as a first step in investigating the mechanism for size control. The choice of this approach is supported by recent theoretical work in which nanoparticle sorting by phase separation was found to be relatively independent of liquid crystal elastic contributions[35].

We represent the isotropic–nematic transition using a Monte Carlo simulation of the Lebwohl–Lasher model[36] in three spatial dimensions. On cooling we observe a first-order phase transition from isotropic to nematic, with nucleation of multiple nematic domains. Next we superimpose a modified Cahn–Hillard (CH) model onto this system[37], where the CH model is adjusted to account for nematic-order-driven phase separation by adding a term to the CH free energy that couples concentration with nematic order. This coupling uses the growth of the nematic order parameter as the driving force for phase separation. As nematic ordering grows, nanoparticle density rises in the isotropic domains and shrinks in the nematic domains (see Methods section). The model specifically focuses on the initial stage of the phase separation process—before the shrinking isotropic domains become dense enough for ligand–ligand interactions to arrest the motion (an effect not treated by our model). We hypothesized that early-stage development of lengthscales in the phase-separation process controls the number of nanoparticles in each assembly, and that these lengthscales influence the outer diameter of the final structures.

Figure 4e shows a simulation snapshot of nanoparticle-rich isotropic domains (red indicates higher nanoparticle concentration) and nanoparticle-poor nematic domains (blue indicates lower nanoparticle concentration). Overlaid arrows are also included to indicate the local nematic director. We can clearly observe that the system has evolved into areas of higher and lower nanoparticle concentrations. Figure 4f shows the corresponding liquid crystal order parameter map for the same system, where yellow indicates area of high-order parameter values and blue corresponds to lower-order parameter values. Side by side, these two images demonstrate a strong correlation between the areas of low liquid crystal order and high particle concentration. These results qualitatively match well with our experimental observations. In Fig. 4g, nanoparticle-rich domain size, as calculated from simulation, is plotted as a function of cooling rate. Domain sizes were determined by spatial correlation function (see Methods section) and averaged over at least five independent simulations. From these data we observed a trend that nanoparticle-rich domain size is linearly related to cooling rate. Representative simulation images are shown inset in Fig. 4g for three different cooling rates. These simulation results support our hypothesis that early nucleation and coarsening lengthscales at the I–N transition point determine the size of the observed nanostructures, producing smaller structures at faster cooling rates.

While this simple model is helpful to understand phase separation in the the first stage of nucleation, it does not account for the dependence of $T_{NI}$ on nanoparticle density, and thus cannot model pattern formation during the second-stage nucleation behavior, which controls the resulting morphology of nanoparticle assemblies. We have also neglected surface

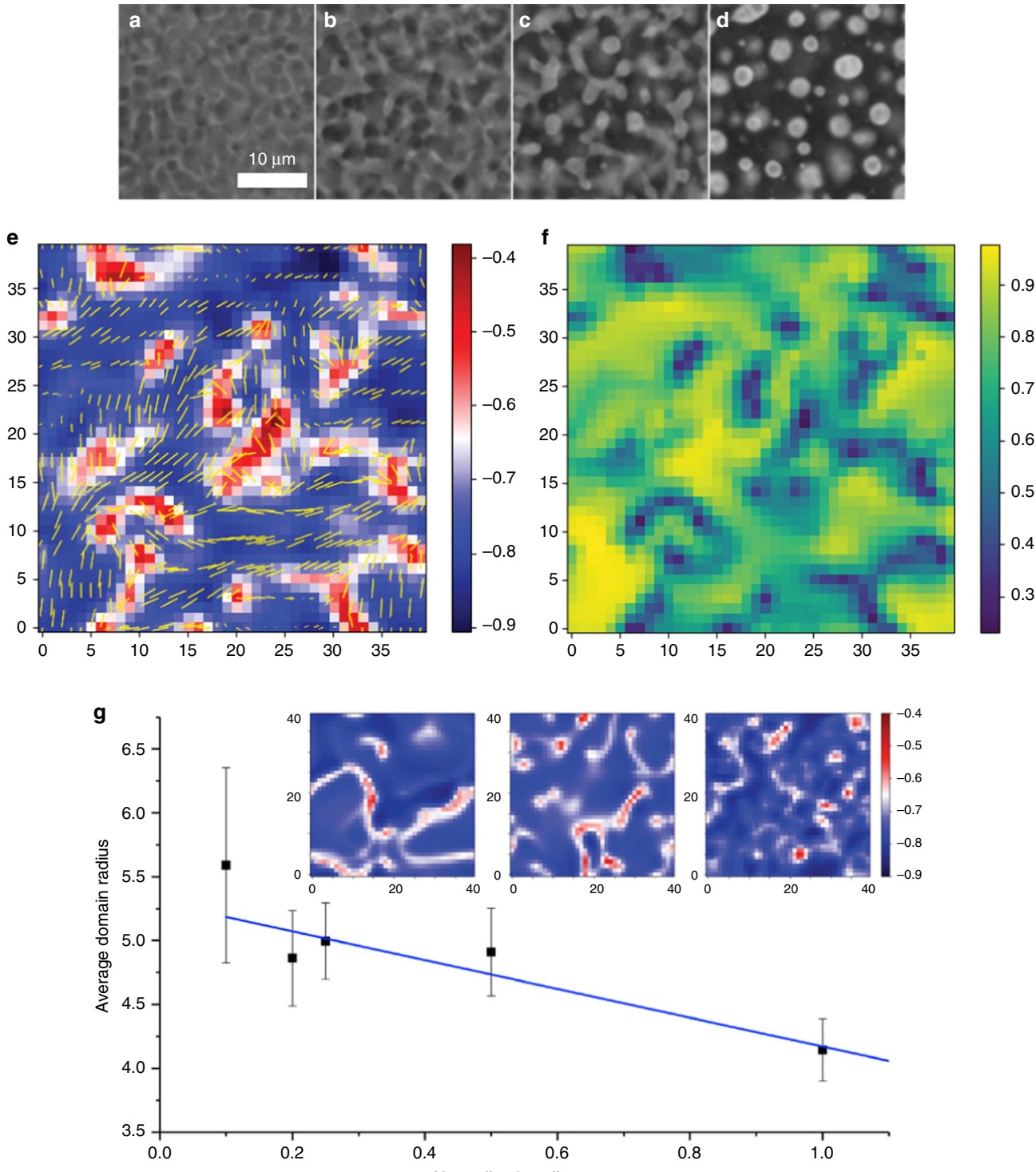

**Fig. 4** Computer simulations of initial stage structure development. **a–d** Experimental high-speed video imaging snapshots representative of four stages in nematic nucleation and growth during the process: **a** initial nucleation and growth, **b** coarsening, **c** domain separation, and **d** secondary nucleation. **e** Two-dimensional (2D) cross-sections of three-dimensional (3D) simulation results for the early stages of nucleation and coarsening showing nanoparticle-rich (redder) and nanoparticle-poor regions (bluer). The images are colored with a linear scale in arbitrary units and overlaid with yellow arrows indicating local director orientation. The concentration distribution is compared with **f**, the calculated order parameter map for the same area using a linear scale from 0 to 1. The comparison clearly indicates a correlation between lower ordered regions and higher particle concentrations. **g** Average particle-rich domain size as a function of cooling rate calculated from simulation results. Error bars represent the standard error of the mean (s.e.m.) measured over five samples. Three simulation snapshots (inset) comparing cooling rates of 0.2, 0.5, and 1.0, from left to right

anchoring effects associated with the boundary between nanoparticles and the nematic liquid crystal. These effects will be considered in future theory/simulation studies.

**Mechanism for structure differentiation.** To better elucidate the governing mechanisms behind network, foam, and capsule formation, we carried out fluorescence microscope video imaging during the formation process. This technique allowed us to track particle distribution throughout the transition—a significant advance over previous related studies. Our observations revealed that the process for all three possible structures can be broken into three stages: (1) particle sorting, in which the initial nematic domains nucleate and grow (concentrating the QDs into shrinking isotropic phase domains), (2) Secondary nematic nucleation, where nematic domains appear inside the shrinking isotropic domains and (3) concentration-induced morphological arrest where particles interact via short range ligand–ligand attraction. As we speculated in a recent publication[28] this arrest likely occurs due to stabilizing intermolecular π interactions between the benzene rings of the ligands as the particles are pushed together. The third step is the key to stabilizing the final structure size, but additional explanation is needed to understand the formation of the three distinct structures under different cooling rates.

Figure 5 shows a time sequence for foam formation (Fig. 5a–f and close up, Fig. 5g). These snapshots reveal an important observation: while the bright isotropic domain is shrinking (but still extended and amorphous in shape), several nematic domains can be seen to simultaneously nucleate and grow inside (Fig. 5g). These secondary domains subsequently grow, leading to multiple inner compartments. This early secondary nucleation is necessary to produce multi-compartment structures. The internal domains grow, pushing the particles together at multiple interior interfaces in a process that produces multiple thin-walled compartments, the particles concentrate and arrest—hence the solid-walled foam-like morphology. We can compare this process to a movie for the formation of single compartment capsules (Supplementary Movie 1). In the case of capsules, we observe that the isotropic domains adopt a spherical shape before the secondary nematic nucleation is observed.

The important question then arises: why do faster cooling rates lead to single compartment shells and slower rates lead to the foam and network morphologies? A critical factor appears to be the timing of secondary nematic nucleation with respect to overall isotropic domain shape evolution. The secondary nematic nucleation drives the particles into their final arrested configurations, but why does this secondary, delayed nucleation occur at all? We can understand this effect simply by considering the effect of impurities on the isotropic-to-nematic phase transition temperature. The nanoparticles in our experiment are ~6 nm in size, a comparable lengthscale to that of the host nematic molecules, and therefore we can consider them as impurities that will depress the I–N transition temperature[26]. In a microscopy study of the I–N phase transition as a function of particle concentration (Supplementary Figure 2), we demonstrate that increasing particle concentration in 5CB from 0.01 to 0.6 wt% depresses the transition temperature by approximately 6 °C. Given the expected higher concentrations of QDs in the shrinking isotropic domains after a quench (compared to the initial low concentration state where the first nematic nucleation occurs), we can expect those interior particle-rich isotropic regions to transition later—provided this step can be considered as analogous to a quasi-static compression with a uniform particle distribution maintained during domain shrinkage. This assumption possibly highlights a key difference between our work and the earlier work by Terentjev and colleagues[23–25] in which

120–150 nm particles were used. In our experiments we were able to image the complete phase separation process, directly verifying uniform particle distribution throughout. Supplementary Figure 1 shows a high-speed video sequence indicating particle equilibration during domain shrinkage and the appearance of the secondary nanoparticle-depleted domain (appearing at ~240 ms). Similar behavior is clearly visible in Supplementary movies 1 and 2.

All three structures begin at the same initial condition, with particles uniformly distributed in the isotropic liquid crystal phase. In the first nucleation stage, nematic domains form and grow, concentrating the particles in the remaining isotropic phase. This stage is demonstrated in Fig. 5b–d for the foam and even more clearly in the detailed time sequence shown in Supplementary Figure 1. Coupled to this process via local particle concentration is the secondary nematic nucleation. This secondary nucleation step ultimately leads to compaction of the particles, arrest, and structure stabilization in either a network, foam, or capsule. Figure 5e, g captures the process as it happens for a structure that ultimately results in the foam morphology. Supplementary Movie 1 shows the process for the capsule morphology. As we can see in both cases, nematic domains form within the bright isotropic domain. We have observed that the relative timing of the two nucleation steps is critical in determining the final arrested structure. If secondary nematic nucleation occurs while the particle-rich isotropic domain is still in a connected network (for example, as represented in Fig. 5i, left, nematic phase shown in grey), the secondary domains will grow and push the particles to form connected tubes. If secondary nucleation occurs at the stage shown in Fig. 5g (large, non-spherical isotropic domains, Fig. 5i, center) we can see multi-compartment foam-like structures. Finally, if secondary nucleation occurs at the spherical domain stage (Fig. 5i, right, Supplementary Movie 1), single compartment shells will likely form.

## Discussion

Growth rates of the primary and secondary nematic nucleations are coupled by an interesting concentration effect. As the primary nematic domains grow and evolve in morphology, simultaneously local particle concentration increases in the isotropic domain. In a fast quench, the nematic phase rapidly nucleates and the system phase separates into many shrinking small spherical isotropic domains, each containing a high concentration of particles. These high local particle concentrations cause the secondary nematic nucleation to occur at a later time since the I–N transition temperature is decreased by several degrees (Supplementary Fig. 2 shows how the I–N phase transition varies as a function of QD concentration). In comparison, a slow quench produces a lower growth exponent for the initial nematic nucleation. This results in larger domains and less rapid concentration of the particles—the secondary nucleation will occur with a shorter time lag, before the system evolves all the way to spherical isotropic domains. Our findings therefore suggest that control of the cooling rate not only provides control over structure size due to coarsening dynamics (as discussed in the previous section), but also provides control over the timing of the secondary nucleation—and thus final morphology (shell, foam, or network).

The reported process of nucleation-and-growth-templating using a liquid crystal phase transition can be compared to the ice-crystal growth templating method used in polymer hydrogel formation[38]. In such systems, the growth of solid crystal domains throughout the material forces dissolved polymers to migrate to the shrinking surrounding fluid phase and porous polymeric hydrogels can be formed. In our liquid crystal system, instead of using a growing solid phase to redistribute dispersed material, we use the nucleation

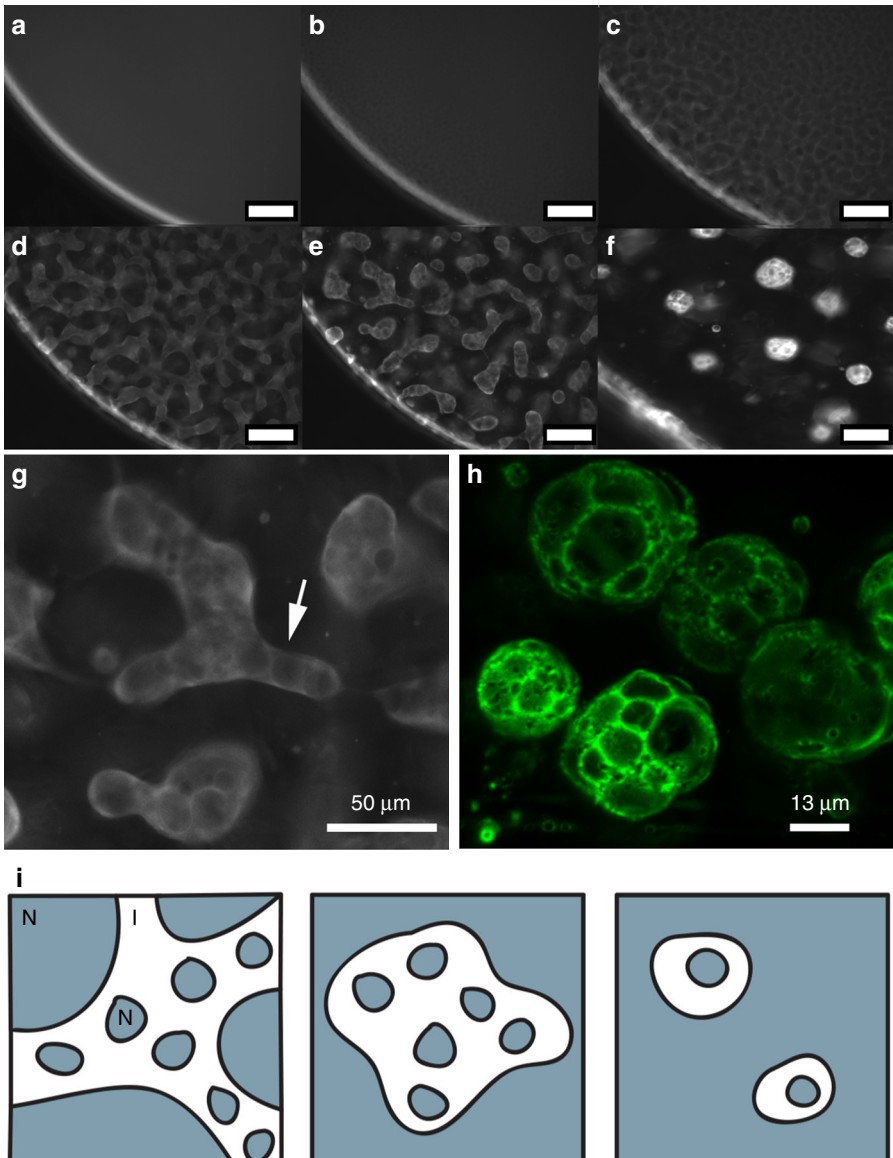

**Fig. 5** Understanding the mechanism for structure differentiation. **a–f** Series of snapshots taken from a fluorescence microscope movie demonstrating the foam formation process in a droplet of liquid crystal with initially well-dispersed quantum dots (QDs). In these images, QD-rich areas appear light and QD-poor areas appear dark. The curved white line in these images is the edge of the droplet, with liquid crystal on the right of this curve. Scale bars for **a–f** are 100 μm. **g** A zoomed-in view from **e** with the arrow indicating inner nematic domain nucleation. **h** A confocal microscope image of several foam structures suspended in 5CB. Images taken from stack of 40 images, Max projected and false colored using Fiji software. **i** schematics illustrating the role of secondary nucleation on final structure showing the particle-rich isotropic phase as white and the nematic phase as gray: left: secondary domains nucleate early on while the isotropic domain is still connected, center: secondary domains nucleate after isotropic domains have separated. and right: secondary domains nucleate late, when isotropic domains have already reached a small size

of the fluid nematic phase to expel nanoparticles, as initially reported in larger particle systems[23,39] and subsequently studied by our group and others[27–29]. In a related experimental system, Yamamoto and Tanaka[40] demonstrated the existence of a second nematic transition using a microemulsion of nanoscale inverse micelles with strong surface anchoring and a biphasic region in the phase diagram. The authors were somewhat speculative on the mechanism but it seems that a phase separation process does result from the growth of nematic domains. Further work will be needed to explore the connection between this work and our own. In our system, particle-induced secondary nematic nucleation provides a route to a rich array of porous structures. The process is not dependent on particle type, and therefore it can be adapted to a

variety of photonic and electromagnetic applications where nanoparticle assembly on the mesoscale is advantageous.

Using two-stage nucleation and growth of nematic domains in an isotropic liquid crystal host phase containing dispersed ligand-modified nanoparticles, we can generate stable nanoparticle superstructures. These structures include extended solid foams and individual capsules ranging from 1 to 50 μm in size. These structures are formed entirely as a response to the isotropic-to-nematic phase transition in a single component liquid crystal host phase with no additional substrate for the nanoparticles or solvent. Our results demonstrate the surprising versatility of this nematic-to-isotropic transition templating approach.

Liquid crystal nanocomposites represent an emerging field with many new phenomena to explore. In particular, structure formation based on domain nucleation and growth in liquid crystals is largely undeveloped. In the liquid crystal family of materials, there is great variation in structure, from the simple nematic phase, to highly complicated structures such as the blue phases, bicontinuous gyroid structures in lyotropic systems, and the many different smectic variant phases. Each of these phases exhibits different growth morphologies related to domain nucleation, and thus we see the potential to generate a large family of LC growth templated structures based on this process.

## Methods

**Ligand synthesis.** Synthesis of 4'-((4-(Octyloxy)benzoyl)oxy)-[1,1'-biphenyl]-4-yl 2-((6-aminohexyl)oxy)-4-(octyloxy)benzoate (**8**): to a 25 mL round bottom flask charged with a PTFE-coated magnetic stir bar was added to a solution of 0.64 g of 2-((6-((*tert*-butoxycarbonyl)amino)hexyl)oxy)-4-(octyloxy)benzoic acid **6** (1.37 mmol) in 7.6 mL of anhydrous toluene. Then, 0.2 mL of thionyl chloride (2.5 mmol) was added dropwise at 0 °C and the reaction was allowed to warm to room temperature and stirred for 24 h. Finally, 0.45 g (1.07 mmol) of **7** was added to the flask and the reaction was heated to 60 °C for 48 h. After concentration under vacuum, purification by flash column chromatography (80:20:00 hexanes/ ethyl acetate/methanol→00:50:50 hexanes/ethyl acetate/methanol on Et₃N-treated SiO₂) afforded **8** (0.330 g, 40%) as a white solid, $R_f$ = 0.89 (50:50 EtOAc:MeOH on an Et₃N-treated SiO₂ TLC plate, visualized by 254 nm light), mp = 94 °C. $^1$H NMR (400 MHz, CDCl₃): δ 8.16 (d, $J$ = 8.8 Hz, 2 H), 8.05 (d, $J$ = 9.1 Hz, 1 H), 7.61 (d, $J$ = 8.6 Hz, 2 H), 7.60 (d, $J$ = 8.4 Hz, 2 H), 7.27 (d, $J$ = 8.6 Hz, 2 H), 7.26 (d, $J$ = 8.4 Hz, 2 H), 6.98 (d, $J$ = 8.7 Hz, 2 H), 6.53 (dd, $J$ = 9.0, 6.5 Hz, 1 H), 6.49 (d, $J$ = 2.5 Hz, 1 H), 4.05 (t, $J$ = 6.5 Hz, 2 H), 4.02 (t, $J$ = 6.5 Hz, 2 H), 4.01 (t, $J$ = 6.5 Hz, 2 H), 2.81 (br s, 2 H), 2.67 (t, $J$ = 6.4 Hz, 2 H), 1.91–1.77 (m, 6 H), 1.57–1.43 (m, 9 H), 1.41–1.26 (m, 17 H), 0.91 (t, $J$ = 6.5 Hz, 3 H), 0.90 (t, $J$ = 6.5 Hz, 3 H); $^{13}$C NMR (125 MHz, CDCl₃): δ 167.9 (C), 167.3 (C), 166.8 (C), 166.3 (C), 164.5 (C), 153.2 (C), 153.1 (C), 140.7 (C), 140.3 (C), 137.1 (CH), 135.0 (2CH), 130.9 (4CH), 125.0 (2CH), 124.8 (2CH), 124.0 (C), 117.0 (2CH), 113.3 (C), 108.2 (CH), 102.8 (CH), 71.4 (CH₂), 71.0 (2CH₂), 42.6 (CH₂), 34.5 (CH₂), 32.0 (2CH₂), 31.9 (2CH₂), 31.8 (CH₂), 31.7 (CH₂), 31.3 (CH₂), 29.7 (CH₂), 28.7 (2CH₂), 28.6 (CH₂), 28.0 (CH₂), 26.8 (CH₂), 25.3 (2CH₂), 16.8 (2CH₃). ATR-FTIR (neat): 2923, 2854, 1726, 1605, 1251, 1196, 1162 cm$^{-1}$. HRMS (ESI) *m/z* calculated for C₄₈H₆₃NO₇ [M]$^+$: 766.4677, found: 766.4659.

**NMR nanoparticle characterization.** $^1$H NMR spectra of purified nanoparticles were collected before and after ligand exchange following established procedure[41]. Using this procedure, we calculated the average ratio of mesogenic ligand to remaining ODA ligand on the particle surface, γ, as the ratio of $X$:$Y$ where $X$ is the area under the triplet corresponding to the mesogenic ligand and $Y$ is the area under the triplet corresponding to the ODA ligand. As before[41], all of the results presented in this paper were produced with a nanoparticle γ ratio of 9:1.

**Nanocomposite preparation.** Different amounts of mesogen-modified QDs in toluene (0.075–0.3 wt%) were mixed with 5CB and bath sonicated at 43 °C (in isotropic phase for the 5CB host) for 3–8 h. This produced a homogeneous dispersion of particles in the liquid crystal and ensured evaporation of any residual toluene. For low QD concentrations (below 0.1 wt%), the amount of added toluene evaporated within 2 h of sonication. Toluene removal was verified by checking the isotropic-to-nematic phase transition temperature. At higher QD concentrations, it was important to test this transition temperature to ensure adequate solvent removal. After dispersion, the mixtures were moved to a 50 °C oven.

To prepare the QD microstructures, clean glass slides and coverslips were coated with an alignment layer to produce the desired LC orientation. For planar alignment (molecules lie parallel to the glass) glass was dip coated with a 1 wt% aqueous polyvinyl alcohol solution, dried, and rubbed with a velvet cloth to induce an alignment direction. The thickness between glass slide and cover slip was tuned to ~120 μm, using a spacer film. All microscope slides were assembled at 50 °C oven to maintain the system's isotropic phase above 34.3 °C. Cooling rate was carefully controlled in these experiments using a Linkham LTS350 hot stage equipped with an in-house designed liquid nitrogen cooled air system. We used several different cooling rates; 7 °C min$^{-1}$, 15 °C min$^{-1}$, 20 °C min$^{-1}$, and 30 °C min$^{-1}$ with this apparatus. In addition, we used a room temperature quenching method to achieve a cooling rate of ~200 °C min$^{-1}$. This was carried out by removing the microscope slide from the hot stage and placing it on a room temperature lab bench. QD fluorescence and the corresponding LC textures were observed using a Leica DM2500P upright microscope equipped with a Q-image Retiga camera. Experiments were performed at three different QD concentrations in liquid crystal; 0.075 wt%, 0.15 wt%, and 0.3 wt%.

**Computer simulation.** To model the initial stage of nanoparticle phase separation, we used a CH model[42,43] representing the phase separation of nanoparticles, coupled to a Lebwohl–Lasher model[36,37] representing the liquid crystal solvent. The Lebwohl–Lasher Hamiltonian is

$$H = \frac{K}{2} \sum_{<a,b>} 1 - \left( \overrightarrow{\mathbf{n^a}} \cdot \overrightarrow{\mathbf{n^b}} \right)^2, \qquad (1)$$

where the spins $\vec{n}^a$ are in a $40 \times 40 \times 40$ cubic lattice and represent the local nematic director at a coarse-grained level, and the sum is over nearest neighbors with periodic boundary conditions. Here $K$ represents a single Frank elastic constant, equal for splay, twist, and bend deformations of the nematic director. To evolve the system forward in time, we calculate the local torque on each spin due to its interaction with nearest neighbors, set the angular velocity $\vec{\omega}$ proportional to the torque, and evolve the spins forward in time via $\vec{n}_{t+1} = \vec{n}_t + (\vec{\omega} \times \vec{n}_t) \Delta t$. This dynamic scheme was selected, rather than a conventional Monte Carlo method, to provide a more realistic time evolution of the director field. Temperature is controlled via a Langevin thermostat. At each time step, the local nematic scalar order parameter $S$ at each lattice site is evaluated by finding the largest eigenvalue of the $Q$-tensor:

$$Q_{i,j} = n^i n^j - \frac{1}{3} \delta_{i,j}, \qquad (2)$$

averaged over a radius of 2 lattice spacings.

On this system, we superimposed a CH model to describe the phase separation process in a binary mixture, in this case a nanoparticle-rich isotropic phase and a nanoparticle-poor nematic phase. The CH equation is given by[42,43]

$$\frac{\partial c}{\partial t} = \nabla^2 (c^3 - c - \nabla^2 c), \qquad (3)$$

where $c$ = $+1$ and $-1$ represent fully separated domains rich in nanoparticles and in liquid crystal, respectively. To this standard form we add a driving term $S$ that represents the scalar order parameter of the liquid crystal:

$$\frac{\partial c}{\partial t} = \nabla^2 (c^3 - c - \nabla^2 c + S). \qquad (4)$$

This new term arises from the addition of a free energy term of the CH equation that is proportional to $cS$, which couples concentration to nematic order. In this way, $S(r,t)$ acts as a dynamic energy landscape, such that rising values of $S$ drive the nanoparticle density away from areas of high orientational order.

We note that this coupling only occurs one way: the presence of an order parameter in the LC drives the diffusion of nanoparticles away from the nematic regions, yet the location of the nanoparticles does not have any effect on the alignment of the LC. This model thus focuses only on how nanoparticle phase separation is affected by the evolving liquid crystal microstructure.

At each time step, we update the nematic director field at all lattice sites; recalculate the local scalar order parameter at all lattice sites; and then integrate the equation of motion for the nanoparticle concentration at all lattice sites. We ran the simulation through the isotropic-to-nematic transition at several different cooling rates. Figure 4e shows a typical two-dimensional slice from an incompletely relaxed three-dimensional nematic director field, showing regions of nematic alignment with a population of $+1/2$ and $-1/2$ defects characteristic of the nematic phase.

To quantify the size of the nanoparticle-rich domains that form during the phase transition, we calculate the spatial correlation function of the density $c(r)$,

$$g(r) = <c(r)c(0)> - <c>^2, \qquad (5)$$

to analyze the final state of each simulation run. We locate the first zero in the correlation function to identify a characteristic lengthscale in the spatial distribution of nanoparticle density. This procedure is carried out by identifying $g(r)$ values that first go from positive to negative, and fitting those points with a fourth-order polynomial. The zeros of this polynomial are then found (two real and two complex) and the lowest positive zero is selected to identify the characteristic domain size $R$. This is the value we take as the average domain size. We then obtain the normalized correlation values as a function of $r/R$. The normalization of $g(r)$ allows for a simpler study of the domain distributions as we are comparing the domains to the average size.

We varied the cooling rate of the simulation by changing the number of timesteps of the simulation. We started at a temperature above the isotropic-nematic phase transition (for our system that was $T$ = 2.0), and cooled to a temperature below the transition. The number of simulation steps ranged from 1000 timesteps for the fastest cooling rate to 10,000 timesteps for the slowest cooling rate. Results for each cooling rate were averaged over 5 runs.

## Data availability

The datasets generated and analyzed during the current study are presented in this published article (and its Supplementary Information) in aggregated form as figures/ graphs and are available from the corresponding author upon reasonable request.

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

## Acknowledgements

The authors would like to acknowledge generous funding from the National Science Foundation grant numbers CBET 1507551, DMR 1409658 and CMMI-1663041. The University of California, Merced, the Merced Nanomaterials Center for Energy and Sensing (funded by NASA grant no. NNX15AQ01A) and an award from the University of California Cancer Research Coordinating committee (CRCC).

## Author contributions

S.T.R. and A.E. performed the microscopy experiments. S.T.R. analyzed the microscopy data. B.J.S., A.K., and G.I.W. designed and carried out chemical synthesis and performed chemical characterization with S.T.R. C.N.M. and R.L.B.S. designed and carried out the computer simulation and analysis. L.S.H. directed the study. S.T.R., L.S.H., B.J.S. and R.L.B.S. wrote the paper.

## Additional information

**Competing interests:** The authors declare no competing interests.

