## [Peer Review File · Nature Communications]

Reviewers' comments:

Reviewer #1 (Remarks to the Author):

This is an interesting work on nematic dispersions of quantum dots, where novel micrometer structures of phase separated QDs are reported. In particular, the authors report the observation of organization of QDs in network-tubular-like morphology, single hollow spheres and foam-like structures. The appearance of these structures depends on the rate of cooling through the isotropic-nematic phase transition and the concentration of QDs. The key hypothesis for the appearance of these hollow microstructures is a new, two stage "phase transition" which involves both the appearance of the nematic order/interface and the migration of QDs into the isotropic regions. This hypothesis is quite reasonable and should be better proved. The experiments are accompanied by numerical modeling of the quench of QDs nematic dispersions.

In principle, this work shows enough novelty and originality to be considered for publication in Nature Communications, after a number of questions are clarified. The work is interesting, because it implies QDs as a potential photonic material, the phase separation of nanoparticles and thermal quench/order nucleation, which is quite fundamental. In spite of novelty I am sceptic regarding the potential for application.

My main concern is related to the justification of the hypothesis of two-stage nucleation. The authors propose that the QDs are driven into the isotropic phase after the nematic is nucleated. Because the concentration of QDs is so high, that it depresses the nematic order in the QD dense region, which appears at much lower temperatures. While this is quite reasonable hypothesis, I would like to see a clear proof for this two stage transition. The authors actually try to describe this in Figures 5(a-f), but the text is so scarce that it is not convincing at all. For example, it is not clear, what is the bright curved line in panel Figure 5a? What is on the left side and what is on the right side of this line? What is the scale in this image?

Furthermore, the manuscript is not well written. In some parts it goes back and forth, repeating some things, passing very fast through important points and is not clear enough in discussing images and what is on particular image. It should be substantially revised to improve the quality of presentation.

From the technical aspect I have the following comments:

1. I propose to measure the "two stage" transition and nucleation process in better detail and make the hypothesis convincing. We have been witnessing great confusion because of experimental errors with different sorts of NPs in nematic LC in the past and would like to have very clear proof of this two stage transition.
2. The authors claim that interfaces do not play any important role for particle aggregation, which surprises me. Why is that so, why the QDs prefer the isotropic phase and not the I-N interface?
3. On page #4, "the solubility" in the isotropic phase is mentioned. What is the "solubility in the isotropic phase? In line 93 it is mentioned that ligands promote this solubility and enhance the particle interactions. It seems to me in great contradiction.
4. Reference #14 is not correctly cited in line #106, this work is not related to particle interaction at interfaces.
5. I am worried about the ligand (substance "8") stability, how well is this molecule attached to the ODA layer at QDs surfaces? Could it diffuse away and mix with 5CB? Any evidence it does not?

6. What is the proof that ligand "8" enhances "dispersability and provides attractive interaction", as claimed in lines 159-160?

7. The scales are missing in panels of Figure 3. Colour scale is not there, time in snapshots a-d is not given, the scale in the insets to panel (g) are not there.

8. Scales are missing in panels (a-g) in Figure 5.

Reviewer #2 (Remarks to the Author):

This paper presents a novel way to make cellular or network structures which are very interesting to a various disciplinary. It utilizes an interesting observation that the particle suppresses the transition temperature of the I-N transitions. This thermodynamic control of the transition (which lead to secondary nucleation), plus the kinetic control of the transition by cooling rate, a rich structure formation is realized. I recommend accept for publication.

The following a few minor points for the authors to make corrections.

(1) Page 9, fig 2(g). It is very hard for me to understand the pattern in the circle. It is the "shading"? How did it stand for NP concentration?

(2) Page 12, Fig.3 (f), what is the color stand for?

(3) Page 14, line 319. The use of "we" is very strange to me. [15] is not the author's paper. Anything wrong here?

(4) "Hollow" in the title is kind of strange. The structure is now full of Nematic phase. How can you wash out the LC molecules, especially for the close spheres (inside the shells)?

Reviewer #3 (Remarks to the Author):

This is an interesting experimental work showing how a nematic liquid crystalline mixture with soluble nanoparticles could be made to spontaneously aggregate into intricate structures. The authors are right to point out that the earlier work on this type of system has used slightly bigger particles: really small quantum dots are used here, and here lies the difference (and the novelty).

I don't fully agree on one aspect of the authors' claim. In one of these 2001 papers [7] it clearly says that the small nanoparticles are the key for this sort of self-assembly by phase separation, and seems to explain that even the sub-micron particles of [7] were in this regime of weak non-topological distortion (as certainly the small quantum dots are as well).

So the only real difference (and novelty) here is the claim of the two-step phase transition with consecutive phase separation of nanoparticles. Fundamentally, this is not impossible: indeed, after the first phase separation the particle-rich isotropic region *could* phase separate again at a lower temperature. In practice, I think the point of [7] was that their primary phase diagram was so wide that the particle-rich isotropic region was already at a concentration where the particles were jammed into a solid (and then, of course, no further thermal equilibration would occur). None of this is discussed here. The authors show many interesting correlations (shell size vs. cooling rate and concentration) but don't offer any reasons or explanation. Nor do I find any actual evidence for the claim of two-step phase separation, except that the fluorescent dots seem to be structured in the interfaces.

In the end of this paper there is a theoretical bit, where the authors explain how they construct the Cahn-Hilliard model for the nanoparticle density with a coupling to the nematic order. Here I don't agree with the authors: the coupling equation (4) has no justification, and certainly differs

strongly from how one would normally expect the nematic order to couple with impurity density (e.g. written in [7], but in fact going way back to De Gennes). The authors need to give at least some justification for their analysis.

In summary: there is no doubt that the authors see very interesting aggregation and self-ordering effects, and observe very interesting laws and correlations. However, the paper as presented has deficiencies and needs to be revised in a big way before a decision of publication in such a prestigious journal could be reached.

Please find below our detailed responses to the different questions and comments from the referees. We have extensively revised the text and figures following their detailed feedback. In addition we have added several supplemental figures and movies to strengthen this work.

Reviewer #1

In principle, this work shows enough novelty and originality to be considered for publication in Nature Communications, after a number of questions are clarified. The work is interesting, because it implies QDs as a potential photonic material, the phase separation of nanoparticles and thermal quench/order nucleation, which is quite fundamental. In spite of novelty I am sceptic regarding the potential for application.

We greatly appreciate that the reviewer considers our work of fundamental interest and suitable for publication in Nature Communications. It's true that our ideas for application are quite speculative, however we feel that is interesting to the reader to propose new ideas for application to stimulate future work on these materials.

My main concern is related to the justification of the hypothesis of two-stage nucleation. The authors propose that the QDs are driven into the isotropic phase after the nematic is nucleated. Because the concentration of QDs is so high, that it depresses the nematic order in the QD dense region, which appears at much lower temperatures. While this is quite reasonable hypothesis, I would like to see a clear proof for this two stage transition. The authors actually try to describe this in Figures 5(a-f), but the text is so scarce that it is not convincing at all. For example, it is not clear, what is the bright curved line in panel Figure 5a? What is on the left side and what is on the right side of this line? What is the scale in this image?

We are grateful to the author for this feedback and have sought to more clearly show the two-stage transition process with added supplemental images, video and description in the text. We have added supplemental materials to support our observations.

To start we addressed the figure clarity issues. The scale bar for panels a-f was in the bottom corner of (a) and applied to all six panels but it was too small. We updated the figure with a larger scale bar in all panels. The curved line is the edge of the material droplet in these figures. To the left of the curve there is empty space (appears uniformly black), whereas to the right our material is shown. QD dense areas appear bright and QD poor areas appear dark. The boundary (curve) of the droplet appears bright because some QD are at the air/LC interface. To clarify this we have improved the figure caption.

This text was added to the caption

"Series of snapshots taken from a fluorescence microscope movie demonstrating the foam formation process in a droplet of liquid crystal with initially well dispersed quantum dots (QDs). In

these images QD rich areas appear light and QD poor areas appear dark. The curved white line in these images is the edge of the droplet, with liquid crystal on the right of this curve"

To show the formation process and appearance of the secondary nematic domains in more detail we have also added a new supplemental figure. This figure shows images taken every 20 ms for the complete process of capsule formation.

The new images clearly show that 1) as the bright QD rich isotropic domains shrink, the particles remain uniformly distributed and then 2) when the second nucleation occurs and a darker inner nematic domain appears, the particles are pushed to the I/N interface. We have also included a supplemental movie showing the formation process for a single capsule, in which inner domains are observed to form and merge inside a spherical domain, ultimately pushing the particles to the outer shell.

The section describing Figure 5 has been significantly modified to remove repetition and improve clarity of description.

In addition we have added a supplemental figure where we plot the relationship between particle concentration in the isotropic phase and the I-N phase transition temperature. As expected, and consistent with Ref 7, we observe a decrease in transition temperature of several degrees if particle concentration in the liquid crystal is increased up to 6x. This further supports our hypothesis that the secondary nucleation is driven by a concentration effect as particles are collected into the shrinking isotropic domains.

Furthermore, the manuscript is not well written. In some parts it goes back and forth, repeating some things, passing very fast through important points and is not clear enough in discussing images and what is on particular image. It should be substantially revised to improve the quality of presentation.

We have made substantial revisions to the text throughout to improve explanations, increase callouts to the figures, and to describe the process and figures more clearly. We have also changed the order of the figures (switching Figures 3 and 4) to prevent jumping "back and forth" in the discussion section.

From the technical aspect I have the following comments:

1. I propose to measure the "two stage" transition and nucleation process in better detail and make the hypothesis convincing. We have been witnessing great confusion because of experimental errors with different sorts of NPs in nematic LC in the past and would like to have very clear proof of this two stage transition.

As described above we have made significant efforts to show the two stage transition behavior clearly. Our supplemental graph of I-N transition temperature as a function of concentration confirms feasibility of the hypothesis by demonstrating that the particles are

capable of inducing significant change in the transition temperature at concentrations relevant to our experiments.

In addition we have included a detailed series of images of capsule formation, and a movie in which the second nucleation is clearly seen appearing inside a spherical isotropic domain. Nematic domains (dark) nucleate and merge, pushing the QDs to the edge of the domain.

2. The authors claim that interfaces do not play any important role for particle aggregation, which surprises me. Why is that so, why the QDs prefer the isotropic phase and not the I-N interface?

We do not in general observe a particular attraction between the QDs and the I/N interface during the phase separation process. Occasionally particle concentration (implied by fluorescence brightness) is slightly enhanced at this interface (although usually it is not) and we presume this is due to the motion of the sweeping phase transition front where particles can become slightly stacked at the boundary (similar to a non-quasistatic compression effect). In colloidal systems researchers have observed larger particles to be attracted to LC defects in the nematic phase or the interface between two different liquids, but these systems differ from ours - we are looking at two phase coexistence of I and N. At this boundary we do not expect to see a particular attraction.

3. On page #4, "the solubility" in the isotropic phase is mentioned. What is the "solubility in the isotropic phase? In line 93 it is mentioned that ligands promote this solubility and enhance the particle interactions. It seems to me in great contradiction.

By solubility we are referring to the ability to the particles to disperse uniformly without aggregating and precipitating out of the phase. In a recent paper (ref 30) our group reported that these particles can be dispersed in the isotropic phase of 5CB although they tend to aggregate in the nematic phase. This behavior was explained by the use of the flexible mesogenic ligands (without these ligands we saw aggregation in the isotropic phase in that paper). We do not however quantify this parameter, but such behavior is determined by observations of uniform fluorescence in the composite material. Figure 5a shows an example of the dispersed state as typically imaged on the fluorescence microscope.

In the process described in this paper we see no contradiction in the statement made above, because particle dispersion in the isotropic phase occurs prior to the aggregate-forming transition to the nematic phase. Particle dispersion in the isotropic phase is facilitated by the improved solubility of the calamitic ligand-functionalized nanoparticles in the 5CB host medium. After the nematic transition the particles are pushed into a close packing where pi-pi interactions between ligands act over just a few angstroms to stabilize the structure; this aromatic interaction is not expected to come into play if the particles are dilutely dispersed in the isotropic phase.

4. Reference #14 is not correctly cited in line #106, this work is not related to particle interaction at interfaces.

The reference has been corrected.

5. I am worried about the ligand (substance "8") stability, how well is this molecule attached to the ODA layer at QDs surfaces? Could it diffuse away and mix with 5CB? Any evidence it does not?

Considering the short timescale of these reactions (less than 1s), leaching is unlikely. Leaching would be about as likely to occur during the ligand exchange we use to install the ligands which takes place over a much longer time period. In addition, the mesostructures are stable to heating up to >100 C without change in structure, thus leaching is not expected to occur to an extent significant enough to change the shells/foams/networks once they've formed. We have seen no evidence of leaching over long time periods (weeks) when reexamining capsules under the microscope.

6. What is the proof that ligand "8" enhances "dispersability and provides attractive interaction", as claimed in lines 159-160?

As described in our answer to query 3, we reported previously (ref 30) the effects of this ligand exchange on QD dispersibility in the isotropic and nematic phases. We have added this reference at that point in the manuscript.

We know that the new ligand 8 enhances thermal stability of the structures when compared to aggregates formed using ODA ligands (our starting point in the ligand exchange)—the structures can be heated up to 120degC before breaking apart, whereas ODA ligand aggregates redisperse at the liquid crystal clearing point. This heating study is published in ref 12.

7. The scales are missing in panels of Figure 3. Colour scale is not there, time in snapshots a-d is not given, the scale in the insets to panel (g) are not there.

This figure has been corrected for scales as requested and the panels improved to show the process more clearly with time stamps.

8. Scales are missing in panels (a-g) in Figure 5.

Scales have been added to the figure.

Reviewer #2 (Remarks to the Author):

This paper presents a novel way to make cellular or network structures which are very interesting to a various disciplinary. It utilizes an interesting observation that the particle suppresses the transition temperature of the I-N transitions. This thermodynamic control of the transition (which lead to secondary nucleation), plus the kinetic control of the transition by cooling rate, a rich structure formation is realized. I recommend accept for publication.

We are extremely pleased that the reviewer recommends acceptance and have carefully addressed the points listed below,

The following a few minor points for the authors to make corrections.

(1) Page 9, fig 2(g). It is very hard for me to understand the pattern in the circle. It is the "shading"? How did it stand for NP concentration?

We have removed the shading effect which was not intended to indicate concentration and agree that it was confusing. The caption has been modified accordingly.

(2) Page 12, Fig.3 (f), what is the color stand for?

This figure has been corrected to show the meaning of the color scales for concentration and order parameter in the different simulation images. The caption has been modified to include more description of the figure.

(3) Page 14, line 319. The use of "we" is very strange to me. [15] is not the author's paper. Anything wrong here?

The reference was incorrect and has been corrected

(4) "Hollow" in the title is kind of strange. The structure is now full of Nematic phase. How can you wash out the LC molecules, especially for the close spheres (inside the shells)?

The word Hollow is used to indicate that the nanoparticle structures formed are not solid, they have spaces inside, we do not intent to imply that there is only air inside. Of course after formation the inner regions are filled with liquid crystal but our preliminary work has shown that this may be removeable as the structures are dried. Hollow also indicates that something can be enclosed inside (i.e. for delivery or encapsulation). The nanoparticle structure itself is hollow can be created with a variety of fillings.

Reviewer #3 (Remarks to the Author):

This is an interesting experimental work showing how a nematic liquid crystalline mixture with soluble nanoparticles could be made to spontaneously aggregate into intricate structures. The authors are right to point out that the earlier work on this type of system has used slightly bigger particles: really small quantum dots are used here, and here lies the difference (and the novelty).

I don't fully agree on one aspect of the authors' claim. In one of these 2001 papers [7] it clearly says that the small nanoparticles are the key for this sort of self-assembly by phase separation, and seems to explain that even the sub-micron particles of [7] were in this regime of weak non-topological distortion (as certainly the small quantum dots are as well).

We are grateful to the reviewer for their careful review and have made some significant revisions to the manuscript to improve clarity and discussion of our arguments, including supplemental figures and movies that support our hypothesis. We agree that our previous QD work and the work in ref 7 should both be in the regime of weak non-topological distortion, but our new system in fact should be even more so: the flexible ligands in our system promote weak surface anchoring, and the QD particles are approximately 15 times smaller in radius.

It's true that the use of small particles in ref 7 is essential to see assembly. The most important difference between their work and our paper, however, may still originate from size differences, specifically the ability of the particles to diffuse in the isotropic phase. Our imaging work suggests that the particles remain in quasi-equilibrium as the isotropic domain shrinks. This hypothesis is supported by images showing a uniform particle distribution throughout the process before the second nucleation occurs. (Supplemental Fig 1 shows it quite well).

Such a condition is less likely possible for a particle 120 nm in radius (considering a simple Stokes-Einstein model of diffusion). If we consider a diffusion length of 1 micron, and estimate diffusion time to be $l^2/6D$ where D is the diffusion constant, using a viscosity of 24 mPa.s at 34°C in the isotropic phase, we find that t for a 3 nm radius particle is just 0.05 s, whereas the equivalent for the 60 nm radius particle is 1.1s. These results strongly suggest that our particles will have time to equilibrate during the "compression" over the observed ~0.5 s time interval (Supplemental fig 1).

We see this as a key difference between the two systems and the main factor contributing to the secondary nucleation effect.

In reference 7 and the following publications the phase separation process and stable two-phase coexistence are not directly experimentally observed. Only the final state showing a cellular structure is imaged and corresponding theory presented to support the hypothesis. In our work we can clearly see the process taking place.

So the only real difference (and novelty) here is the claim of the two-step phase transition with consecutive phase separation of nanoparticles. Fundamentally, this is not impossible: indeed, after the first phase separation the particle-rich isotropic region *could* phase separate again at a lower temperature. In practice, I think the point of [7] was that their primary phase diagram was so wide that the particle-rich isotropic region was already at a concentration where the particles were jammed into a solid (and then, of course, no further thermal equilibration would occur). None of this is discussed here.

An important difference between our work and that of [7] is that we have imaged the phase transition process using fluorescence high speed video capture. This means we are able to see the particle distribution throughout the process. The particles in [7] are ~120nm in size and significantly more concentrated in the initial state, therefore it is not too surprising that they might jam at an earlier time. The mechanism discussed in [7] hypothesizes that the particles excluded by the growing nematic domains are jammed together, at the stage where our system is still in the bicontinuous network state. Here the size difference is important - our imaging reveals a slightly different mechanism in which the system passes through the bicontinuous network point, the isotropic domains pinch off, and continue to shrink, minimizing their interfaces.

An additional important factor related to particle size comes from the particles' ability to remain in equilibrium during the formation process - this is verified by examining images to see that the fluorescence in the isotropic domains is uniform. This effect was also reported by us earlier for different ligands in a system that did not display the secondary nucleation [ref 11].

To show the formation process and appearance of the secondary nematic domains in more detail we have added a new supplemental figure showing images taken every 20ms for the complete process of capsule formation.

The new image (S1) clearly shows 1) as the bright QD rich Isotropic domains shrink, the particles remain uniformly distributed and in thermal equilibrium and then 2) movies 1,2 show that when the second nucleation occurs and a darker inner nematic domain appears, the particles are pushed to the I/N interface. Supplemental movie 1 shows the formation process for a single capsule, in which inner domains are observed to form and merge inside a spherical domain, ultimately pushing the particles to the outer shell.

In addition we have added another supplemental figure where we plot the relationship between particle concentration in the isotropic phase and the I-N phase transition temperature. As expected, and consistent with a similar plot in Ref 7, we observe a decrease in transition temperature of several degrees if particle concentration in the liquid crystal is increased up to 6x. This further supports our hypothesis that the secondary nucleation is driven by a concentration effect as particles are collected into the shrinking isotropic domains.

The authors show many interesting correlations (shell size vs. cooling rate and concentration) but don't offer any reasons or explanation.

The key finding of our paper is that a second nematic phase nucleation takes place in the shrinking isotropic domains during phase separation and that this secondary nucleation occurs due to a concentration effect. The timing of the secondary nucleation with respect to isotropic domain shape determines the final structure.

We did spend some time in the manuscript considering mechanisms for size control, and this was one of the main aims of the simulation (to plot pinched-off domain size as a function of cooling rate). Ultimately this is a subject for more detailed future investigation as there are a variety of coupled contributions to be taken into account and further theoretical insights will be very valuable.

Nor do I find any actual evidence for the claim of two-step phase separation, except that the fluorescent dots seem to be structured in the interfaces.

See our answer to the point above regarding additional supplemental materials. Figure 5g also clearly shows the appearance of secondary nematic domains within a shrinking isotropic domain. They appear dark as particles are expelled.

In the end of this paper there is a theoretical bit, where the authors explain how they construct the Cahn-Hilliard model for the nanoparticle density with a coupling to the nematic order. Here I don't agree with the authors: the coupling equation (4) has no justification, and certainly differs strongly from how one would normally expect the nematic order to couple with impurity density (e.g. written in [7], but in fact going way back to De Gennes). The authors need to give at least some justification for their analysis.

In our simulation study we aimed to determine approximately how the dynamics of the isotropic-nematic transition results in the segregation of nanoparticles into isolated clusters and their resulting average mass as a function of cooling rate. Rather than attempt a fully coupled model as in [7], we carried out a far simpler calculation.

First, we ran a simulation study of the isotropic-nematic transition on cooling, WITHOUT any nanoparticles, using the Lebwohl-Lasher model with director relaxation dynamics. The resulting microstructural evolution shows nucleation and growth of nematic domains. At some point, the nematic domains percolate and the remaining isotropic domains become isolated, with nematic on all sides.

Second, we made the assumption that even in a liquid crystal WITH nanoparticles, the nucleation and growth of nematic domains would follow roughly the same trajectory on

cooling, up to that point. (Only after that would the high concentration of nanoparticles in the isotropic domains significantly affect the dynamics of the phase transition.)

Third, we ran a simulation of the Cahn-Hilliard model for nanoparticle phase separation. We added a term to the free energy proportional to cS , where c represents the concentration of nanoparticles and S is the local scalar order parameter of the liquid crystal. We used $S(r,t)$ from our Lebwohl-Lasher simulation as a dynamic potential energy landscape, such that as the scalar order parameter rises in the growing nematic domains, nanoparticles are driven out and segregate to the isotropic domains. When the isotropic domains eventually become isolated, at that point we stop and calculate total mass of nanoparticles collected in each isotropic domain, and calculate the resulting mass distribution. We ran the simulation at different cooling rates and calculated the average of the resulting mass distribution for each.

This simplified simulation approach thus has only one-way coupling: the liquid crystal's microstructure evolves independently as if it were pure, while the nanoparticle density evolves in response to the $S(r,t)$ landscape from the liquid crystal.

This model can only predict the mass of nanoparticles segregated to each isotropic domain at the moment it becomes isolated. It cannot model the microstructural evolution that happens on further cooling. This simple simulation did however show that because of the relatively high diffusivity of nanoparticles, their density is roughly uniform in the isotropic domains when those domains first become isolated.

In order to model the subsequent microstructural evolution, we would need to reformulate our simulation approach to add (1) coupling between the two models in both directions, rather than just one direction, and (2) a model for the anchoring energy of the nematic liquid crystal in contact with nanoparticle aggregates. This is beyond the scope of the current work but will be an interesting topic of study in the future, and should allow interesting comparisons to the theoretical analysis in reference [7].

Reviewers' comments:

Reviewer #1 (Remarks to the Author):

This is now a great manuscript, very clear and informative, with good narration. The authors have made considerable effort and succeeded in improving the manuscript. I recommend publication in Nature Communications.

Reviewer #3 (Remarks to the Author):

The article remains intriguing, but in my opinion - not ready for a high-profile journal. In their Reply, the authors essentially re-stated the questions posed to them, but didn't answer them (except one: an illustration that very small QDs would be able to equilibrate during the phase-separation process). Theoretical model used for their simulation is poor, and the excuse that it is "simple" isn't really valid. The added images, although quite attractive, still don't provide any evidence for a sequence of two phase transitions. Having seen the revisions and the Reply, I don't see how could the authors improve this article - and therefore recommend it sent to a specialized journal.

In the paper we reported the formation of new mesoscale structures (foams, capsules and networks), consisting of nanoparticles stabilized by interacting of mesogenic ligands. A key finding of our work is in the new mechanism for structure formation, which we describe as a double nematic nucleation. A secondary nematic nucleation occurs, driven by increased particle concentration in shrinking particle-rich isotropic domains. This secondary nucleation results in the formation of a variety of novel size-tunable solid porous structures composed of nanoparticles.

In this letter I discuss only the comments received from reviewer 3 from both of their reviews. Reviewers 1 and 2 recommended acceptance of the paper in its current form.

In the first round of comments, reviewer 3 wrote several comments which we answered in some detail. Here I will summarize/quote these initial comments (**bold**) and our responses:

They began:

“This is an interesting experimental work showing how a nematic liquid crystalline mixture with soluble nanoparticles could be made to spontaneously aggregate into intricate structures. The authors are right to point out that the earlier work on this type of system has used slightly bigger particles: really small quantum dots are used here, and here lies the difference (and the novelty). I don't fully agree on one aspect of the authors' claim. In one of these 2001 papers [7] it clearly says that the small nanoparticles are the key for this sort of self-assembly by phase separation, and seems to explain that even the sub-micron particles of [7] were in this regime of weak non-topological distortion (as certainly the small quantum dots are as well).

So the only real difference (and novelty) here is the claim of the two-step phase transition with consecutive phase separation of nanoparticles. Fundamentally, this is not impossible: indeed, after the first phase separation the particle-rich isotropic region *could* phase separate again at a lower temperature. In practice, I think the point of [7] was that their primary phase diagram was so wide that the particle-rich isotropic region was already at a concentration where the particles were jammed into a solid (and then, of course, no further thermal equilibration would occur). None of this is discussed here.”

The reviewer establishes that they think the results are interesting and then focuses on the relationship between our work and one of the references [7]. Their points are that a) the particles in [7] are also small and then (b) that they believe the double nematic nucleation could be possible but they note that in reference [7] this did not occur, presumably due to earlier stage jamming of the particles.

The particles in [7] are actually quite a bit larger than ours ~120nm (we use 6nm QDs) and significantly more concentrated in the initial state than ours, therefore it is not too surprising that they might jam at an earlier time. The mechanism discussed in [7] hypothesizes that the particles excluded by the growing nematic domains jam in the early phase separation stage – at the point where our system is still in the bicontinuous network state (QD-poor nematic and QD-rich isotropic). Here the size difference is very important - our direct imaging reveals that our system passes through this bicontinuous network point, and particle-rich isotopic domains pinch off, continuing to shrink and concentrate the particles.

We responded to this comment by adding additional imaging directly showing the formation process.

- a) We added supplemental images (a high-speed camera time series of capsule formation) and two supplemental movies directly imaging QD distribution during the formation process. Our aim was to demonstrate that very small particles allow the shrinking isotropic domains to relax to a quasi-equilibrium state during the cooling process. This observation is key to our hypothesis in which we propose that a second nematic nucleation takes place inside the shrinking domains – leading to the porous structures.
- b) We discussed the possibility for this “equilibrium” condition in our response in detail considering particle sizes and its feasibility comparing our work with that published in [7], plus we added text to the paper in support of this discussion.
- c) In addition we added new data - we plotted the I-N phase transition temperature as a function of particle concentration in a bulk isotropic phase. As expected, and consistent with a similar plot in [7], we observed a decrease in transition temperature of several degrees if particle concentration in the liquid crystal is increased up to 6x. This further supported our hypothesis that the secondary nucleation is driven by a concentration effect as particles are collected into the shrinking isotropic domains.

In their second response the reviewer accepted that we had made “an illustration that very small QDs would be able to equilibrate during the phase-separation process”

It should be noted that the formation process was not imaged or studied experimentally in reference 7. That paper discusses a related system, but the particles are larger and they only image the final state – there is no observation of how the system gets to that state.

By contrast, our experiments use fluorescence microscopy to directly image particle distribution throughout the formation process. This allows us to see the distribution of nanoparticles (quantum dots) whereas the liquid crystal is not visible. It is very clear that this method can be used to indicate the spatial organization of the particles as the system passes through the I-N phase transition and the technique has been well established by our group (A.L. Rodarte, J Mat Chem 2013, ChemPhysChem 2014).

The new image (S1) shows that, 1) as the bright QD rich Isotropic domains shrink, the particles tend to remain uniformly distributed and then 2) when the second nucleation occurs a darker inner nematic domain appears, the particles are only then pushed to the I/N interface. The supplemental movies show the formation process for a single capsule and multiple capsules simultaneously, in which inner domains are observed to form and merge inside a spherical domain, ultimately pushing the particles to the outer shell.

Another comment by the reviewer from their original review was

“The authors show many interesting correlations (shell size vs. cooling rate and concentration) but don't offer any reasons or explanation.”

In fact, we did spend time in the manuscript discussing ideas regarding the dependence of shell size on cooling rate and concentration. We proposed that the timing of the secondary nucleation with respect to isotropic domain shape determines the final structure.

We also discussed mechanisms for size control, in particular

- a) Cooling rate – We investigated length-scales in the system at the point where isotropic domains pinched off from the shrinking bicontinuous network and their dependence on cooling rate.

- b) Initial particle concentration – We fitted size data as function of capsule size to a simple formula describing the compression of enclosed particles in a sphere onto a 2D thin shell.

Another comment was,

“Nor do I find any actual evidence for the claim of two-step phase separation, except that the fluorescent dots seem to be structured in the interfaces.”

This comment seemed to indicate that we had perhaps not provided enough imaging data – thus we added the additional movies and image series.

If we consider that the reviewer in a later comment accepted that the quasi-equilibration was observed in the time series (supplemental image) and combine that conclusion with the observation of late nucleation and growth of a dark domain inside the isotropic domains (seen in the figures and movies) we believe that there is good evidence for the proposed process.

Finally, the reviewer made objections to the simulations in the paper

“In the end of this paper there is a theoretical bit, where the authors explain how they construct the Cahn-Hilliard model for the nanoparticle density with a coupling to the nematic order. Here I don't agree with the authors: the coupling equation (4) has no justification, and certainly differs strongly from how one would normally expect the nematic order to couple with impurity density (e.g. written in [7], but in fact going way back to De Gennes). The authors need to give at least some justification for their analysis.”

In our original response we explained the rationale behind this simple simulation as an exploration of the development of phase separated length-scales as a function of cooling rate. The idea was to support our hypotheses on control of shell size by cooling rate.

We aimed to determine approximately how the dynamics of the isotropic-nematic transition results in the segregation of nanoparticles into isolated clusters and their resulting average mass (i.e. domain size) as a function of cooling rate.

First, we ran a simulation study of the isotropic-nematic transition on cooling, WITHOUT any nanoparticles, using the Lebwohl-Lasher model with director relaxation dynamics. The resulting microstructural evolution shows nucleation and growth of nematic domains. At some point, the nematic domains percolate and the remaining isotropic domains become isolated, with nematic on all sides. – This is the key moment we were interested in looking at. When the isotropic domains separate – all the contained particles in those domains go on to become shells.

We made the assumption that even in a liquid crystal WITH nanoparticles, the nucleation and growth of nematic domains would follow roughly the same trajectory on cooling, up to that point. (Only after that would the high concentration of nanoparticles in the isotropic domains significantly affect the dynamics of the phase transition.). Our assumption is valid if we assume that the initial particle concentration is dilute (an important difference between our work and ref 7).

Third, we ran a simulation of the Cahn-Hilliard model for nanoparticle phase separation. We added a term to the free energy proportional to cS , where c represents the concentration of nanoparticles and S is the local scalar order parameter of the liquid crystal. We used $S(r,t)$ from our Lebwohl-Lasher simulation as a dynamic potential energy landscape, such that as the scalar order parameter rises in the growing nematic domains, nanoparticles are driven out and segregate to the isotropic

domains. When the isotropic domains eventually become isolated, at that point we stop and calculate total mass of nanoparticles collected in each isotropic domain, and calculate the resulting mass distribution. We ran the simulation at different cooling rates and calculated the average of the resulting mass distribution for each.

This simplified simulation approach thus has only one-way coupling: the liquid crystal's microstructure evolves independently as if it were pure, while the nanoparticle density evolves in response to the $S(r,t)$ landscape from the liquid crystal.

We want to emphasize that this model can only predict the mass of nanoparticles segregated to each isotropic domain at the moment it becomes isolated. It cannot model the microstructural evolution that happens on further cooling. This was not our intention and we agree that a full model of microstructural evolution, would need a more detailed simulation approach including anchoring energy between particles and nematic LC. This however was not the intention of our work.

Since submitting this work, a collaborator published a relevant paper which also supports our simplified approach (now ref 35).

In this work they model the growth of a nematic domain inside an isotropic region containing nanoparticles, looking at how the particles are expelled from the growing nematic domain in a spherical symmetry. In [7] the authors speculate on the role of topological defects as a stabilization mechanism for their system, and include Frank elasticity in their model. However according to the work by the Guzman group the presence of topological defects is not expected to play an important role in structure formation and stabilization (stabilization occurs due to our ligands) in our system. This theoretical work by the Guzman group (Using a full Landau-DeGennes and elastic free energy theory) supports our computational approach because it demonstrates that nanoparticle sorting by phase separation is relatively independent of liquid crystal elastic contributions and topological defect configuration and that nematic order parameter is the key driving factor.

In their second response, the reviewer 3 made some more general comments,

1. “The article remains intriguing, but in my opinion - not ready for a high-profile journal.”

The reviewer says the paper is “not ready” for a high profile journal, but does not explain why. This does not form a logical argument or valid reason for rejection.

2. “In their Reply, the authors essentially re-stated the questions posed to them, but didn't answer them (except one: an illustration that very small QDs would be able to equilibrate during the phase-separation process).”

This is simply untrue, we discussed all comments made by this reviewer (and others) extensively in our response. Reviewer 3 in fact did not ask any specific questions in their original review, so we did not understand this comment. Most of their initial comments were regarding lack of detail or proof of the proposed concepts and we believe that these have now been addressed adequately. The reviewer spent some time referring to a specific paper (reference 7). While this is an important and relevant reference we believe that we have, at this point made adequate reference and discussion of that earlier work – including emphasizing its importance.

3. “Theoretical model used for their simulation is poor, and the excuse that it is “simple” isn't really valid.”

It is unfortunate that the reviewer did not accept the utility of our simplified approach but we believe that for the purposes it was intended, in the dilute limit, the simulation is valid. In addition, as mentioned above, since the original submission, a new paper by our collaborator Guzman supports our approach where we neglect anchoring energies due to the particle surfaces.

4. “The added images, although quite attractive, still don't provide any evidence for a sequence of two phase transitions.”

We strongly disagree with this statement. It does not make sense for the reviewer to write “don't provide any evidence”.

The nematic phase appears as darker areas in our fluorescence images as the particles are expelled from that phase. Our figure 5g and supplemental movies and supplemental figure (capturing a timed sequence) clearly show the appearance of dark domains inside the structures at later times after a period of thermal equilibration. It might be claimed that our original figure image of the foam inner domain formation does not show sufficient temporal detail. For this reason, we added the high-speed capture sequence of the nucleation process and movies.

We also added clarifying text on this point to the paper

“In our experiments we were able to image the complete phase separation process, directly verifying uniform particle distribution throughout. Supplemental Figure S1 shows a high-speed video sequence indicating particle equilibration during domain shrinkage and the appearance of the secondary nanoparticle-depleted domain (appearing at ~240ms). Similar behavior is clearly visible in the supplemental movies.”

In summary, to prove the double nematic nucleation effect we have provided

1. Direct imaging of particle distribution during the initial phase transition process showing clear particle sorting by phase.
2. Direct imaging of thermal equilibration of the particles within the shrinking domain (a key point which the reviewer accepted).
3. Direct imaging of the secondary nucleation appearing within the isotropic domains (dark patches). This imaging was challenging considering the temporal resolution and small structure size.
4. A plot showing the effect of particle concentration on the I-N phase transition.

If considered appropriate we can also include an additional more quantitative analysis of the inner domain formation using intensity line cuts on the images, additional arrows to images and any other clarifications to guide the reader's eye as might be suggested by the reviewers.

Reviewers' comments:

Reviewer #3 (Remarks to the Author):

It appears I am the only "obstacle" for this paper, and I certainly don't wish to fight it. The authors seem to really want to publish it in the present form (their Reply is long and forceful), so may be they should.

My comments do not change very much:

1. The theoretical model they used in their simulation is poor, but it is known that lots of models of evolving nucleation can produce these 'spinodal-looking' structures - as their have. I don't understand why they need this model in the paper.

(Their argument that the Guzman paper [35] supports theirs isn't reasonable: if anyone (ref.[7] or others) has claimed that topological defects play a role for small particles - they were simply wrong, but that doesn't justify using a model where the coupling $c \cdot S$ is used in the Hamiltonian).

2. The particles are "small" when they are below the topological defect limit (which is a few hundred nm). But the particles here are much smaller, so perhaps the fact they are below the nematic coherence length (which is about 10nm) is relevant. For example Tanaka had a paper (Nature, v.409, 321, 2001) which showed intriguing differences when such small particles are in the nematic.

3. There is no doubt that small particles would go to interfaces (or disclination cores) of the growing nematic phase: there are very many papers about that, all showing such concentration in different ways. The issue here is about the second nucleation. On re-reading their paper and Reply, I suspect they mean the effect that is expressed in Fig.1 of Tanaka's paper mentioned above (the two-stage isotropic-nematic transition). I don't understand why they don't show any DSC data on very slow cooling to support this. Perhaps the difference in optical images (which is all this paper has) is that Tanaka had a thick cell, and different type of particles?

Reviewer #4 (Remarks to the Author):

I have carefully evaluated the aspects raised in the disputed between the authors and referee 3. I can confirm that the evidence and novelty of the secondary nucleation approach is covered comprehensively, and that I find this aspect novel and well described.

Reviewer #4

“Reviewer #4 was not asked to review the manuscript in full, but in separate comments to the editor, he/she made one suggestion: he/she believes the phase diagram shown in Fig 2g is oversimplified and would prefer that it is described as qualitative phase behavior.”

We have corrected the Figure 2 caption and some places in the text to clearly indicate that this is a qualitative phase diagram/qualitative phase behavior.

Reviewer #3:

This reviewer made three specific comments, these are answered separately below,

“It appears I am the only "obstacle" for this paper, and I certainly don't wish to fight it. The authors seem to really want to publish it in the present form (their Reply is long and forceful), so may be they should. My comments do not change very much:

1. The theoretical model they used in their simulation is poor, but it is known that lots of models of evolving nucleation can produce these 'spinodal-looking' structures - as their have. I don't understand why they need this model in the paper.

(Their argument that the Guzman paper [35] supports theirs isn't reasonable: if anyone (ref.[7] or others) has claimed that topological defects play a role for small particles - they were simply wrong, but that doesn't justify using a model where the coupling c^*S is used in the Hamiltonian)."

We have made efforts throughout the paper to moderate our language regarding the simulation and emphasize the simplified approach, which is intended only to represent the first stage of nucleation, and to emphasize that results are qualitative. This has been made clear now in a revised abstract and text.

Notably we added

“Our aim was to model only the first stage of the phase separation process and see if the lengthscale/cooling rate dependence could qualitatively be recovered as a first step in investigating the mechanism for size control.”

The following paragraph was also revised somewhat to be more clear.

At the end of this section we added the following to address your request for caveats of the model.

“While this simple model is helpful to understand phase separation in the first stage of nucleation, it does not account for the dependence of T_{NI} on nanoparticle density, and thus cannot model pattern formation during the second stage nucleation behavior, which controls the resulting morphology of nanoparticle assemblies. We have also neglected surface-anchoring effects associated with the boundary between nanoparticles and the nematic liquid crystal. These effects will be considered in future theory/simulation studies.”

Some small changes to the methods section 4.3 “computer simulation” were also added for clarification.

“2. The particles are "small" when they are below the topological defect limit (which is a few hundred nm). But the particles here are much smaller, so perhaps the fact they are below the nematic coherence length (which is about 10nm) is relevant. For example Tanaka had a paper (Nature, v.409, 321, 2001) which showed intriguing differences when such small particles are in the nematic.”

We do believe that the small size of our particles (6nm) is very relevant to the observed behavior and it will be interesting in future to look at the behavior of even smaller systems. The suggested paper by Tanaka an important addition to the picture makes a good addition to our references. I have added it with a short comment in the discussion section. The text there now reads

“In a related experimental system, Yamamoto and Tanaka [40] demonstrated the existence of a second nematic transition using a microemulsion of nanoscale inverse micelles with strong surface anchoring and a biphasic region in the phase diagram. The authors were somewhat speculative on the mechanism but it seems that a phase separation process does result from the growth nematic domains. Further work will be needed to explore the connection between this work and our own. In our system, particle-induced secondary nematic nucleation provides a novel route to a rich array of porous structures. The process is not dependent on particle type, therefore it can be adapted to a variety of

photonic and electromagnetic applications where nanoparticle assembly on the mesoscale is advantageous.”

Reference numbers have been updated accordingly.

“3. There is no doubt that small particles would go to interfaces (or disclination cores) of the growing nematic phase: there are very many papers about that, all showing such concentration in different ways. The issue here is about the second nucleation. On re-reading their paper and Reply, I suspect they mean the effect that is expressed in Fig.1 of Tanaka's paper mentioned above (the two-stage isotropic-nematic transition). I don't understand why they don't show any DSC data on very slow cooling to support this. Perhaps the difference in optical images (which is all this paper has) is that Tanaka had a thick cell, and different type of particles?”

In the Tanaka paper (also see comment above), they use slow cooling (0.3deg/min) to record a double DSC peak. They report biphasic behavior with stable two-phase coexistence. In our system the behavior is a little less clear-cut since the conditions to form the porous structures only occur at high cooling rates (7-200deg/min). At low cooling rates we tend to generate very large domains and phase separation comparable to the cell size, thus we do not see formation of the reported structures (capsules, foams etc) and have not yet focused in detail on that regime. It will be very interesting to investigate the phase behavior at different cooling rates further using DSC in a future study.